# Enhanced nanochannel translocation and localization of genomic DNA molecules using three-dimensional nanofunnels

Jinsheng Zhou [1], Yanqian Wang[2], Laurent D. Menard[1], Sergey Panyukov [3], Michael Rubinstein[1,4] & J. Michael Ramsey[1,4,5,6]

The ability to precisely control the transport of single DNA molecules through a nanoscale channel is critical to DNA sequencing and mapping technologies that are currently under development. Here we show how the electrokinetically driven introduction of DNA molecules into a nanochannel is facilitated by incorporating a three-dimensional nanofunnel at the nanochannel entrance. Individual DNA molecules are imaged as they attempt to overcome the entropic barrier to nanochannel entry through nanofunnels with various shapes. Theoretical modeling of this behavior reveals the pushing and pulling forces that result in up to a 30-fold reduction in the threshold electric field needed to initiate nanochannel entry. In some cases, DNA molecules are stably trapped and axially positioned within a nanofunnel at sub-threshold electric field strengths, suggesting the utility of nanofunnels as force spectroscopy tools. These applications illustrate the benefit of finely tuning nanoscale conduit geometries, which can be designed using the theoretical model developed here.

[1] Department of Chemistry, University of North Carolina, Chapel Hill, NC 27599, USA. [2] Department of Physics and Astronomy, University of North Carolina, Chapel Hill, NC 27599, USA. [3] PN Lebedev Physical Institute, Russian Academy of Sciences, Moscow 117924, Russia. [4] Department of Applied Physical Sciences, University of North Carolina, Chapel Hill, NC 27599, USA. [5] Department of Biomedical Engineering, University of North Carolina, Chapel Hill, NC 27599, USA. [6] Carolina Center for Genome Sciences, University of North Carolina, Chapel Hill, NC 27599, USA. Correspondence and requests for materials should be addressed to J.M.R. (email: jmramsey@unc.edu)

Threading a macromolecule such as genomic DNA through a nanopore or nanochannel forces its extension and ensures the sequential passage of molecular segments through a nanoscale volume. Electrical or optical probing of this volume produces a highly localized signal that can be correlated to the structure or nucleotide sequence of the DNA[1–5]. The transport of DNA molecules through nanoscale conduits is most often achieved by applying an electric field across the conduit, which induces an electrostatic force on the negatively charged DNA and pulls it into the confines of the nanopore or nanochannel. This force must be sufficient to overcome the free-energy barrier to DNA entry into a nanopore or nanochannel that results from the reduced conformational entropy of the confined macromolecule[6, 7]. The geometry of the region where critical dimensions decrease from the microscale to the nanoscale has been found to strongly affect the dynamics of this process in nanofluidic platforms based on biological pore complexes or channels fabricated in insulating substrates[4, 8–11]. Control over transport dynamics in turn affects the throughput and resolving power of such platforms vis-à-vis the efficiency with which DNA molecules are introduced to the nanoscale region and the speed with which the DNA passes through the detection volume. Despite the acknowledged importance of geometry on performance, however, it is difficult to develop a detailed understanding of its role in the nanofluidic platforms thus far reported. For example, gradient structures consisting of an array of posts aligned to a nanochannel array were described by Cao et al.[8], where the inter-post distance and the channel depth decrease from the DNA reservoir to the nanochannel entrances. Multiple pathways of gradually increasing confinement are thus provided to a molecule. Although these structures are demonstrably useful for facilitating DNA entry into the nanochannels, their multi-path nature complicates the study of DNA behavior, which has not been modeled therein. The direct-write fabrication method of focused ion beam (FIB) milling can be used to pattern nanofluidic structures in a substrate with control over both their width and depth.

Here we describe the fabrication of nanochannels having three-dimensional nanofunnel entrances of various shapes using FIB-milling, visualization of DNA behavior in these nanofunnels, and modeling of this behavior to better understand how controlling the geometry of the nanochannel entrance can enhance the electrokinetic manipulation of DNA molecules in nanofluidic platforms. We note that a FIB-milled "funnel-like inlet," consisting of a series of discrete reductions in conduit width and depth, has previously been reported to assist with DNA entry into a nanochannel, although the effect was not quantified[11]. In the present study, the gradual and smooth transition from microscale to nanoscale confinement within our FIB-milled nanofunnels, in contrast to a coarser transitioning of DNA confinement in a stepwise fashion, is an important enabling aspect of both the experimental measurements and modeling efforts.

## Results

**Model of electrokinetic DNA entry into a nanochannel.** In the absence of a nanofunnel (Fig. 1a), the high electric field in the nanochannel acts only on those nucleotides closest to the entrance, which pull the entire DNA molecule into the nanochannel if the force is sufficient to overcome the opposing entropic force[6, 7, 12]. Although the frequency of DNA threading can be increased by applying a larger voltage, this approach is problematic if the resulting DNA transport velocity exceeds the sampling rates of electronic or optical detection modes[13–19]. By incorporating a three-dimensional nanofunnel at the nanochannel entrance (Fig. 1b), DNA can be more efficiently introduced into the nanochannel without an increase in the nanochannel electric field. The nanofunnel can be shaped such that at its mouth the electrohydrodynamic (coupled electrostatic and hydrodynamic) force gradient is greater than the entropic force gradient and DNA entry into the nanofunnel is unimpeded. As the net force acting on the DNA molecule within the nanofunnel drives it towards the nanochannel entrance, the increasing confinement partially extends the DNA molecule, reducing its conformational entropy. The forces in the nanofunnel have thus done some work on the DNA molecule and the molecule is in a conformation that can more easily enter the nanochannel than in the case without a nanofunnel. The presence of the nanofunnel furthermore generates an additional force acting on the entire DNA molecule that assists in pushing it into the nanochannel entrance, an osmotic gradient force that arises from variation in the DNA monomer concentration along the longitudinal axis of the nanofunnel. The contributions of these forces are shown schematically in Fig. 1, where the arrows of different colors suggest an average force over all monomers at a particular location within the nanofunnel. We note that the osmotic gradient and entropic forces are acting only on monomers in contact with each other or with the nanofunnel wall. As the DNA molecule fluctuates, these forces are directly applied onto different monomers at different moments in

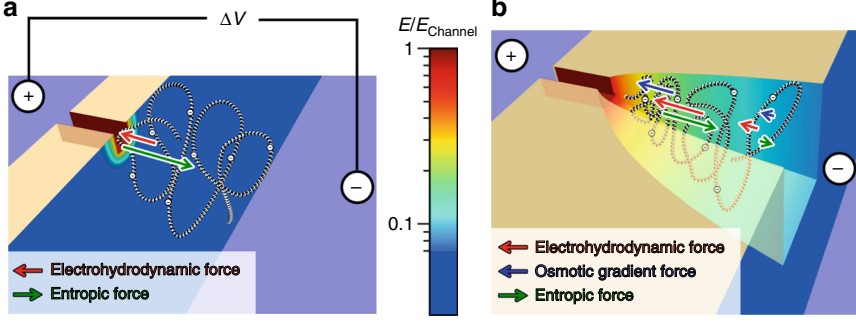

**Fig. 1** Effect of a nanofunnel on DNA threading into a nanochannel. **a** Cartoon illustrating the electrohydrodynamic (red arrow) and entropic (green arrow) forces affecting the leading portion of the DNA molecule as it is electrophoretically pulled into a nanochannel under an applied bias, $\Delta V$. **b** These forces and an additional osmotic gradient force (blue arrows) act on the entire DNA molecule as a result of the extended high electric field region and the confining effects of the nanofunnel (in contrast to the smaller portion of segments affected in (**a**)). At the wide entrance to the nanofunnel (its mouth), the electrohydrodynamic force is greater than the entropic force. The osmotic gradient force acts as an "electro-osmotic piston," providing further assistance for DNA entry into the nanochannel. The electric field strengths (relative to the electric field strength in the nanochannel, $E_{Channel}$) are indicated using a logarithmic color scale for clarity

time. The concentration fluctuations are suppressed within the mean field theory, and the DNA configuration is completely characterized by the forces pre-averaged over all monomers at a location along the nanofunnel's longitudinal axis. Our theoretical model shows, and experimental measurements confirmed, that the forces experienced by the DNA molecule in the nanofunnel result in compression of the DNA leading sections towards the nanochannel entrance by DNA trailing sections, an effect that we refer to as an electro-osmotic piston.

**DNA behavior imaged in three-dimensional nanofunnels.** We imaged individual fluorescently-stained λ-phage and T4-phage genomic DNA molecules (stained contour lengths of 21 and 72 μm, respectively) as they were electrokinetically driven towards a nanochannel through a three-dimensional nanofunnel having a square cross-sectional profile that decreased gradually in both width and depth from 1600 to 120 nm (Fig. 2a; Supplementary Figs. 1–3). These dimensions range from ones on the order of the

molecules' radii of gyration (0.6 μm and 1.3 μm for λ-phage and T4-phage DNA, respectively)[20] to length scales approaching double-stranded DNA's persistence length (~50 nm)[21], the decay distance of chain orientational correlation along the molecule. When the electric field was sufficiently high, a molecule would immediately enter the nanochannel upon reaching the nanofunnel–nanochannel junction (Fig. 2b, red line i). At intermediate electric fields, a molecule would reside for a time, τ, at the nanochannel entrance, repeatedly attempting to overcome the free-energy barrier before successfully entering (Fig. 2b, blue line ii, τ = 47 s). At low electric fields, the residence time increased so that a DNA molecule was sustainably trapped within the nanofunnel (Fig. 2b, green line iii). The low and intermediate field behaviors were modeled theoretically by balancing the confinement, electrohydrodynamic, and osmotic gradient forces acting on each of the sections of a DNA molecule. In addition to the identification of the electro-osmotic piston, a significant outcome of these modeling efforts is an extension of the important work by Long, Viovy, and Ajdari[22] to describe the action of electric fields and non-electric forces on DNA molecules in confined environments. The positional dependence of the forces can be represented by an effective free-energy profile of the DNA molecule (Fig. 2c) with the difference between the energy minimum and maximum corresponding to the effective free-energy barrier (ΔF) to nanochannel entry (see Supplementary Eqs. 1–18 for discussion on the development of effective free energy Supplementary Eq. 19). Increasing the electric field reduces the barrier height, decreasing the residence time according to the Arrhenius relation:

$$\tau \cong \tau_0 e^{\Delta F / k_B T} \qquad (1)$$

where $\tau_0$ is the minimum time needed for the molecule's leading sections to diffusively enter the nanochannel[23, 24], $k_B$ is Boltzmann's constant, and $T$ is absolute temperature. At electric field strengths where $\Delta F$ is sufficiently greater than $k_B T$, the free-energy minimum corresponds to the trapping position of

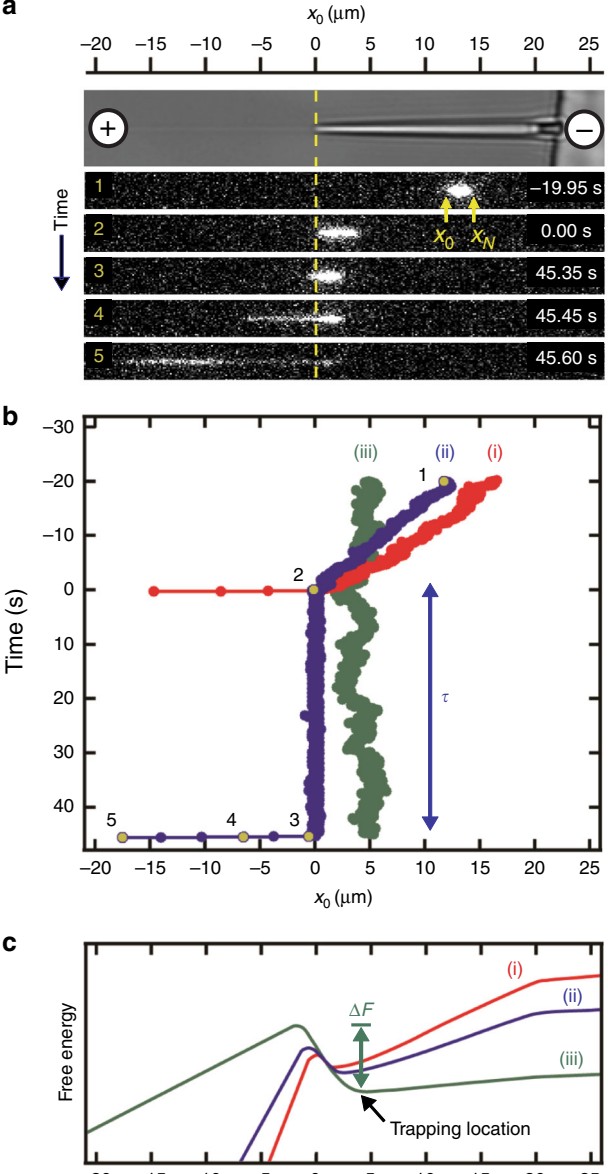

**Fig. 2** Measurement of DNA molecules within a three-dimensional nanofunnel. **a** Representative images recording the position and conformation of a λ-phage DNA molecule at various time points as it is electrokinetically driven from right to left through a nanofunnel and into the associated nanochannel. The top panel is a bright-field image showing the position of the nanofunnel (positive x coordinates) and nanochannel (negative x coordinates) and the voltage polarity applied across the nanofunnel–nanochannel device. The numbered frames (1–5) are fluorescence images of the DNA molecule stained with an intercalating dye recorded at the indicated time points. Image analysis determined the positions of the molecule's leading ($x_0$) and trailing ($x_N$) ends at each time point. **b** The position of a molecule's leading end within a nanofunnel measured at three different nanochannel electric field strengths: 77.5 V cm⁻¹ (red line, i), 54.3 V cm⁻¹ (blue line, ii), and 15.5 V cm⁻¹ (green line, iii). The blue line (ii) represents behavior at an intermediate electric field strength, where the DNA molecule has a finite residence time (τ = 47 s) at the nanochannel entrance prior to the entry. The numbered gold circles indicate the leading edge measured from the numbered fluorescence images in (**a**). **c** Relative effective free energies at different nanochannel electric field strengths of a DNA molecule as a function of its leading end position within the nanofunnel–nanochannel. The line colors and labels correspond to the same electric field conditions as in (**b**). As the electric field strength decreases, the energy barrier to nanochannel entry (ΔF) increases and the energy minimum, which corresponds to a trapping location, moves away from the nanochannel entrance towards the nanofunnel mouth

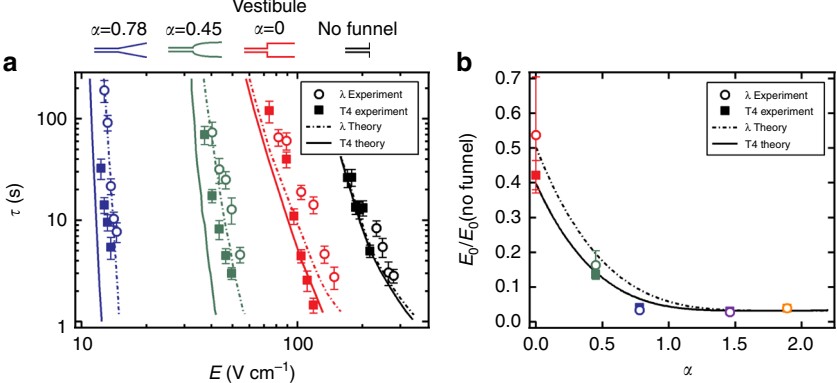

**Fig. 3** Residence time measurements and threshold electric field reduction. **a** Mean residence times measured at various nanochannel electric field strengths, $E$, in three different nanofunnels (defined by $\alpha = 0.78$, $\alpha = 0.45$, and $\alpha = 0$; see Eq. 2) and in the absence of a nanofunnel. Each dataset is associated with the nanofunnel indicated directly above it in the figure and is color coded accordingly. Data were collected using stained λ-phage (48.5 kbp, open circles) and T4-phage (165.6 kbp, filled squares) DNA molecules. The error bars indicate the standard deviations of at least 20 independent measurements per experimental data point. The curves represent the best-fit of the theoretical model to these experimental data. **b** Extrapolating the experimental electric field data in (**a**) to $\tau_0 = 6$ ms results in the characteristic threshold electric field strength, $E_0$ (normalized to the threshold electric field strength measured in the absence of a nanofunnel $E_0$ (no funnel)) for each of the three nanofunnels (red, green, and blue symbols obtained from the data in (**a**) of the same colors). $\tau_0 = 6$ ms corresponds to the Zimm relaxation time of the leading portion (~3000 bp) of the nanofunnel-confined molecule[23, 24]. This 3000-bp segment is the length of DNA that must enter the nanochannel to initiate threading, as determined from the theoretical monomer concentration profiles shown in Supplementary Fig. 7b. The experimental values in (**b**) for the $\alpha = 1.46$ and $\alpha = 1.89$ nanofunnels were determined as described in the Supplementary Methods. The solid and dashed black lines are interpolations of theoretical values calculated for T4-phage and λ-phage DNA, respectively, over a wider range of $\alpha$ values. The error bars indicate the $1\sigma$ confidence level of the threshold electric field strengths determined from the experimental data and are smaller than the symbols for the results from nanofunnels where $\alpha > 0.45$

the DNA molecule located some distance from the nanochannel entrance (Fig. 2c, green line iii).

**Residence times and threshold electric field strengths**. The field-dependent residence times were measured in nanofunnels with comparable dimensions but different shapes, as well as for DNA entry into a nanochannel without an incorporated nanofunnel. The nanofunnel shapes were defined by the following equation:

$$y(x) \approx z(x) = D\left(\frac{x}{x_D}\right)^\alpha \qquad (2)$$

where $y(x)$ and $z(x)$ are the funnel width and depth, respectively, at position $x > 0$ along the funnel's longitudinal axis, $D$ is the widest dimension of the nanofunnel, and $x_D$ is the nanofunnel length. In this study, $D = 1.6 \pm 0.1 \, \mu m$, $x_D = 21.5 \pm 0.2 \, \mu m$, and the nanochannel width and depth were each $120 \pm 15$ nm. Residence time measurements were performed in nanofunnels defined by $\alpha$ values of 0, 0.45, and 0.78, whereas supplementary experiments were conducted in nanofunnels with $\alpha$ values of 1.46 and 1.89 (Supplementary Fig. 4). Residence times were also calculated theoretically by determining the minimum and maximum of effective free energy as a function of DNA end coordinate $x_0$ at each nanochannel electric field strength and then calculating their difference, $\Delta F$, and therefore $\tau$.

We found that in each device with a nanofunnel, the longer T4-phage DNA molecules entered the nanochannel more readily than λ-phage DNA molecules (Fig. 3a). This effect is due to the increased size of the molecule's trailing portion that contributes to a stronger electro-osmotic piston. We note that the $\alpha = 0$ "nanofunnel" (or vestibule) produced a significant reduction in the electric field needed to initiate nanochannel entry, compared with the nanochannels without a nanofunnel, despite a relatively abrupt change in confinement at the nanofunnel–nanochannel junction. This is consistent with the field in the vestibule being greater than that in the device

microchannels, which contributes to greater DNA compression and a stronger electro-osmotic piston. Qualitatively, this finding agrees with the higher rate of single-stranded DNA entry into an α-hemolysin nanopore through the vestibule (*cis*) side of the pore complex compared to the narrow (*trans*) side of the pore[9, 10].

The transition from the $\alpha = 0$ nanofunnel to the nanochannel is similar to an entropic trap such as those reported by Han et al.[25], which consisted of 30-μm wide channels with alternating deep (1.4 μm) and shallow (90 nm) segments. The critical dimensions of our $\alpha = 0$ nanofunnel are similar, but the reduction in both the width and depth dimensions in our case presents both a larger entropic barrier and a smaller collisional cross-section between the DNA molecules and the nanochannel entrance. It is therefore unsurprising that the residence times directly measured at the nanochannel entrance are longer (by an order of magnitude) than those calculated for comparable electric fields by Han et al. from mobility data in entropic trap devices. As in our experiments, Han and Craighead[26] observed shorter residence times for longer DNA molecules, an effect that, when multiplied over thousands of entropic traps in series, enabled size-dependent DNA separations.

As the value of $\alpha$ increased, DNA entry into the nanochannel became easier. The high electric field of the nanochannel extends farther into the nanofunnel as does the region of greater confinement. The increased gradients of the opposing electro-hydrodynamic and entropic forces therefore result in greater compression and a greater osmotic gradient force. We used the nanochannel electric field $E_0$ at which $\Delta F = 0$ to compare the effectiveness of various nanofunnels. As this is the field strength at which $\tau = \tau_0$, values of $E_0$ can be estimated by extrapolating the experimental data in Fig. 3a to short residence times. The relative $E_0$ values thus estimated are shown in Fig. 3b, along with values obtained from the measurements in the $\alpha = 1.46$ and $\alpha = 1.89$ nanofunnels. The greater than 30-fold reduction in $E_0$ that is observed experimentally is in close agreement to the reduction predicted theoretically.

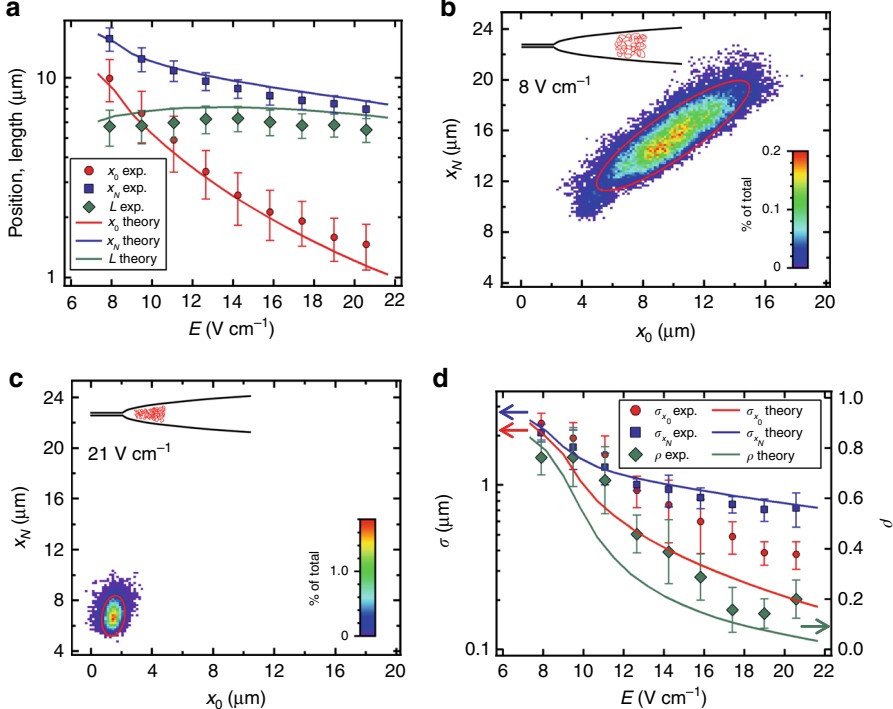

**Fig. 4** Field-dependent positions and fluctuations of trapped T4-phage DNA. **a** Experimentally measured mean values of DNA end positions and length (symbols) at each electric field in an $\alpha = 0.45$ nanofunnel compared with the theoretically predicted values (curves). The error bars are the standard deviations of the multiple independent measurements taken over the entire imaging period at a given electric field strength. **b, c** Filled contour plots showing the probabilities of $x_0$, $x_N$ coordinates measured from each fluorescence image at the low field (8 V cm$^{-1}$) and high field (21 V cm$^{-1}$) conditions of the stable trapping regime, respectively. The color scales indicate the percentage of measurements corresponding to the various $x_0$, $x_N$ pairs. The red ellipses indicate the best-fit bivariate normal distributions ($2\sigma$) to the data. The insets show schematically the DNA conformation (position, length, and packing density) associated with these operating conditions. **d** Comparison of parameters of the bivariate normal distribution fitting analysis of the experimental data (symbols) and theoretical probability distributions (curves). The error bars indicate the $1\sigma$ confidence level of the parameters obtained from the bivariate Gaussian fits. There is a significant decrease in thermal fluctuations at increased field strengths (i.e., stronger trapping). These fluctuations are highly correlated ($\rho \sim 0.8$) at low field strengths as the molecule diffuses as a whole in a soft trapping potential. Similar results for λ-phage DNA are provided in Supplementary Fig. 5

**Stable DNA trapping in a nanofunnel**. Stable trapping was systematically measured in the $\alpha = 0.45$ nanofunnel by imaging a single DNA molecule at several trapping field strengths (Fig. 4; Supplementary Fig. 5). The mean position of the trapped molecule is sensitive to the strength of the applied electric field and we refer to nanofunnels operated in this mode as "electro-osmotic tweezers" by analogy to optical or magnetic tweezers. Figure 4a compares the experimental mean positions of the leading end ($x_0$) and trailing end ($x_N$) of a T4-phage DNA molecule trapped over a range of field strengths to the respective values predicted by theory. As the molecule moves from the nanofunnel mouth to the nanochannel entrance with increasing field strength, its extension length, $= \overline{x_N} - \overline{x_0}$, first increases as a result of the increased confinement but then decreases slightly at higher electric fields as a result of the electro-osmotic compression despite a continued increase in confinement[23]. We note that in these quasi-equilibrium measurements, the DNA molecules are extended by confinement and not as a result of any significant strain rate imposed by the electric field gradient[27].

Varying the applied electric field of the electro-osmotic tweezers also affects the stiffness of the trap and therefore the magnitude of DNA thermal fluctuations. Measured values of various combinations of end coordinates $x_0$, $x_N$ for a T4-phage DNA molecule trapped at the representative nanochannel electric fields of 8 V cm$^{-1}$ and 21 V cm$^{-1}$ are shown in Fig. 4b and c, respectively. Such data were fit to bivariate normal distributions

(red ellipses in Fig. 4b, c) to determine the fluctuations of the end positions ($\sigma_{x_0}$, $\sigma_{x_N}$) and the coefficient of their correlation ($\rho$). These results were compared (Fig. 4d) to values determined theoretically by calculating the minimal work necessary to displace the molecule's ends from their mean positions (see Supplementary Eqs. 20–42 for discussion on the development of thermal fluctuation Supplementary Eqs. 43 and 44 and correlation Supplementary Eq. 45). At the lowest field strengths, the molecule is weakly trapped and thermal fluctuations are larger and highly correlated as the molecule fluctuates as a whole along the longitudinal nanofunnel axis. At higher electric fields, the correlations between fluctuations of the ends are reduced as compression of the leading sections of the molecule suppresses the fluctuations of this end, whereas the less constrained trailing end of the molecule is freer to fluctuate.

Each of the above parameters determined by imaging a stably trapped DNA molecule are sensitive to the length, linear charge density, effective width, and persistence length of the molecule (Supplementary Information). Electro-osmotic tweezers can therefore be used to measure changes to these characteristics that are produced, for example, by changes to the solution ionic strength[28]. We note that the use of multiple complementary force spectroscopies is useful for revealing subtle differences in polymer dynamical behavior[29]. Although DNA stretching and twisting behavior has been extensively studied using atomic force microscopy, optical tweezers, and magnetic tweezers, far fewer measurements of DNA compression have been reported due to a

lack of available tools[30]. Previously reported compression of DNA molecules confined within a nanochannel against an obstruction or nanobead is analogous to compression within an $\alpha = 0$ nanofunnel[31–33]. In the present study, however, we demonstrate electro-osmotic tweezers that are tunable through the nanofunnel dimensions and geometry, promising the availability of measurements to various charged macromolecules across different force regimes. In addition to complementing existing stretching and twisting force spectroscopies, the ability to measure compression is relevant to better understanding phenomena such as nucleic acid packaging (e.g., in viral capsids)[34].

In conclusion, the force gradients experienced by a DNA molecule as it is electrokinetically driven from a microscale reservoir into a nanochannel are highly dependent on the geometry at the nanochannel entrance that defines how abruptly or gradually a DNA molecule experiences increased confinement. Through a combination of experimental and theoretical results, we demonstrated that nanofunnels with a shape defined by $\alpha \approx 1.5$ were most effective at lowering the field strength needed to drive DNA transport through a nanochannel. Given the rapid decrease in relative threshold electric field strengths as a function of $\alpha$, nanofunnels with $\alpha = 0.78$ were nearly as effective. The observation that longer DNA molecules can more readily enter a nanochannel as a result of the nanofunnel and the electro-osmotic piston is an important outcome of this study and contrasts with past findings. Nanofunnels defined by smaller $\alpha$ values were more useful for trapping DNA molecules at voltage-dependent positions within the nanofunnel. Both the residence time and trapping measurements that are the focus of this work describe a system that is near quasi-equilibrium. In fact, the technological utility of the three-dimensional nanofunnels derives in large part from the ability to operate these nanofunnel–nanochannel devices at low electric field strengths. This contrasts from much of the existing work wherein DNA molecules are electrophoretically driven through pores or channels using high electric field strengths to achieve practical device throughput. We expect that the theoretical model of DNA behavior described here will guide future refinements in structures that can provide enhanced transport control in nanofluidic-based nucleic acid and protein analysis platforms[35]. Although the FIB-milling fabrication method used here is admittedly a low-throughput technique, the flexibility of generating highly useful three-dimensional nanofunnels motivates its use, especially in conjunction with higher throughput fabrication methods. FIB-milling could be used, for example, to fabricate masters for the subsequent fabrication of many fluidic devices through molding or embossing methods[11].

## Methods

**Device fabrication**. Nanofluidic devices were fabricated in fused silica substrates coated with a 130-nm thick chromium film. The microfluidic channels that provided fluid and DNA access to the nanofluidic components were patterned using standard photolithography and hydrogen fluoride-based wet etching techniques where typical dimensions were 210 μm wide and 100 μm deep. There was a ~100 μm gap between the independent microchannels, which was spanned by the nanofunnel–nanochannel pair in a subsequent fabrication step. To access the microfluidic channels, vias were drilled from the substrate backside using abrasive powder blasting. The nanofunnel and nanochannel were milled through the chromium film into the underlying fused silica substrate using a focused ion beam (FIB) instrument (Helios NanoLab, FEI Company)[36]. The two components were milled simultaneously using a single gray scale bitmap image to define the geometry in the $x$–$y$ plane (positions over which the beam was scanned) as well as the depth in the $z$ dimension (dwell time of the beam at each $x$, $y$ coordinate). Test structures were milled assuming a linear relationship between pixel intensity and milling depth. The varying nanofunnel width, however, was found to cause deviation from a linear milling rate, likely due to the dependence of sputtering yield on the beam angle of incidence and the impact of feature dimensions on material redeposition[37]. The pixel intensities in the bitmap image were therefore empirically adjusted to compensate for these nonlinear effects,

resulting in a nearly uniform aspect ratio (depth:width) along the entire length of the nanofunnel.

Following FIB-milling, the chromium film was removed using a chemical etchant (Transene Company, Inc.) and a new 10-nm chromium film was deposited using ion beam sputtering (Model IBS/e, South Bay Technologies, Inc.). The purpose of this second film was to dissipate charge during subsequent electron microscopy. The nanofunnel and nanochannel were imaged using scanning electron microscopy (SEM) and atomic force microscopy (AFM) with a high-aspect-ratio probe (ACCESS-NC probe, Applied NanoStructures, Inc.). After imaging, the chromium film was chemically etched, the substrate cleaned by immersion in a stabilized piranha solution (Nanostrip 2X, Cyantek Corporation), and the top surface bonded to a clean fused silica coverslip. Permanent fusion bonding was achieved by heating the bonded substrate to 1000 °C and holding at this temperature for 48 h. After bonding, reservoirs were affixed over the powder blasted vias with a UV-curable epoxy to facilitate the introduction of solutions to the fluidic network.

**Measuring DNA molecules in three-dimensional nanofunnels**. Lambda-phage DNA (Promega) or T4-phage DNA (Nippon Gene) in 2X TBE was stained with the intercalating dye, YOYO-1 (Invitrogen), at a base-pair:dye ratio of 5:1. Solutions containing 0.5 ng μL$^{-1}$ of DNA (16 pM for λ-phage and 4.6 pM for T4-phage) also contained 4% (by volume) β-mercaptoethanol (Fisher Scientific) to limit photo-induced damage and 2% (by mass) polyvinylpyrrolidone (PVP, 10 kDa, Sigma-Aldrich) to reduce electro-osmotic flow within the channels. The device was mounted on an inverted fluorescence microscope (Nikon) and imaged through a ×100/1.4 NA oil immersion objective (Nikon) as DNA molecules were being electrokinetically driven through the microfluidic and nanofluidic channels. Fluorescence was excited using a 100-W mercury arc lamp filtered through a GFP-3035 filter set (Semrock) and video images were recorded using a Cascade II EM-CCD camera (Photometrics) at 20 frames per second. Neutral density filters were inserted into the excitation optical path to reduce the irradiance to 40 mW cm$^{-2}$. Under these conditions, a single molecule could be observed for at least 30 min without any evidence of photo-induced damage.

The DNA concentrations used were sufficiently low that occupancy of two molecules within a nanofunnel during a measurement was rare. Introduction of a molecule to the nanofunnel was therefore expedited by using a relatively high (1500 V cm$^{-1}$) nanochannel electric field to draw in a molecule from the source microfluidic channel (corresponding to an electric field at the nanofunnel mouth of 5–10 V cm$^{-1}$, depending on nanofunnel shape). The voltage was then zeroed momentarily before being reapplied at the magnitude appropriate to generate the desired nanochannel electric field (8–22 V cm$^{-1}$ for trapping experiments and up to 260 V cm$^{-1}$ for residence time measurements).

At electric field strengths sufficient to force a DNA molecule into the nanochannel, the molecule was imaged as it was transported down the nanofunnel towards the nanochannel and as it resided at the nanochannel entrance attempting entry (Fig. 2). Image recording was terminated once the molecule was pulled into the nanochannel. The residence times of at least 20 molecules were measured at each electric field strength. The protocol for measuring residence times of DNA molecules at the entrance of the nanochannel in the absence of a nanofunnel was slightly different. The greater diffusivity of the molecules at the nanochannel entrance in the absence of a nanofunnel made molecule-tracking measurements more ambiguous than illustrated in Fig. 2. To circumvent the difficulty of identifying that a molecule was at the nanochannel entrance, the residence time measurements were conducted by first ensuring that the molecule started immediately proximal to the entrance. This was done by electrokinetically introducing a DNA molecule to the entrance region through the nanochannel from its distal (exit) end. As soon as the DNA molecule was detected in the microchannel near the nanochannel entrance, the voltage was zeroed. Following a programmed delay of 0.5 s for λ-phage DNA or 2 s for T4-phage DNA to allow the molecule to relax, the appropriate voltage was applied to generate the desired nanochannel electric field and the time until the molecule entered the nanochannel was measured.

To record the electro-osmotic tweezing behavior in the $\alpha = 0.45$ nanofunnel, a molecule was imaged at each field strength for at least 10 min This resulted in at least 12,000 images per field that were analyzed to determine the distribution of the fluctuating molecule's positions and extension lengths. In the event that photo-induced damage was observed, the DNA molecule was ejected from the nanofunnel and a new molecule was introduced to continue with the measurements. The $\alpha = 0.45$ nanofunnel was ideal for these measurements because the DNA molecule's position was measurably field-dependent over a relatively wide range of field strengths. Nanofunnels with larger values of $\alpha$ (up to $\alpha \approx 1.5$) exhibited the same trapping capabilities but over a much smaller field range.

**Image analysis**. Images recording DNA position were analyzed using an automated program written in MATLAB to extract the location of the molecule's ends at each time point. These ends experimentally correspond to the most distal and most proximal bright spots in the image and not exactly to the first and last base pairs of the molecule. The end positions in the scaling model are localized only down to the size of the corresponding "blobs"[38]. It is therefore to this level of accuracy that the measured end positions coincide with the positions of the

corresponding terminal bases of the molecule. Images were acquired with the nanofunnel and nanochannel oriented horizontally so the program first identified the highest intensity horizontal line profile. The profile was smoothed using a moving average along the profile and the baseline intensity and highest intensity point along the smoothed profile were determined. The ends of the molecule were identified as the extents of the region of the smoothed intensity profile that exceeded a threshold above baseline that was 20% of the highest intensity point. These coarse positions were then refined by using them as the starting points for identifying the molecule's ends from the raw unsmoothed data. Starting from the initial position, the data were scanned to the high intensity side until a threshold equal to 5× the baseline standard deviation was reached, confirming that the coarse position correctly identified the molecule's edge. Then the data were scanned back to the low-intensity side until the signal was 3× the baseline standard deviation, which indicated the precise position of detectable fluorescence from the DNA molecule. This protocol was highly effective at preventing errors given the relatively low signal-to-noise resulting from the low-intensity excitation used to avoid photo-induced damage.

**Theoretical modeling of DNA trapping.** DNA was modeled as a semi-flexible chain having a persistence length of 50 nm and an effective DNA width of 6 nm that includes the contribution of the double layer formed by counterions around the backbone[39]. Contained within a nanofunnel and in the presence of an electric field, each segment of the DNA molecule is acted upon by the three forces indicated in Fig. 1b. The electrohydrodynamic force, $f_{eh}$, describes the summation of the electrostatic force acting on the polyanionic DNA molecule and the hydrodynamic force induced by the fluid flow in the direction opposite to DNA migration driven by counterions around the DNA backbone. In the Supplementary Discussion, we generalize an analysis of electrohydrodynamic forces given in ref. [22] to the case of DNA trapped inside a nanofunnel, the walls of which suppress the fluid backflow around the molecule. In this case, the backflow can circulate only on microscopic scales through the effective pores inside DNA, similar to the case of a charged gel. The entropic force, $f_{entropic}$, results primarily from the reduction of conformational degrees of freedom of the DNA molecule as it moves deeper into the nanofunnel. The molecule's elasticity that resists its stretching is another contributor to the entropic force that was included in the calculations. For a given segment of the molecule, the opposing electrohydrodynamic and entropic forces are typically unbalanced. This imbalance results in molecular compression, an increase in local monomer density, and an increase in the repulsion (steric and electrostatic) between monomers. The monomer density varies along the longitudinal nanofunnel axis because the magnitudes of the electrohydrodynamic and entropic forces are dependent upon a segment's position within the funnel. This concentration gradient results in an osmotic pressure gradient along the longitudinal nanofunnel axis and an osmotic gradient force, $f_{osm}$.

The equations for each of the three forces described above are developed in the Supplementary Discussion. The conformation of the stably trapped DNA molecule—position, length, and monomer density profile—at each electric field condition was solved by balancing these forces. Equivalently, the contribution of each of the three forces to the molecule's effective free energy was calculated and the quasi-equilibrium conformation was determined by minimizing the effective free energy (Supplementary Eq. 19). Thermal fluctuations around the mean conformation were determined at each electric field by calculating the work, $R_{min}$, necessary to displace the molecule ends over a range of $\delta x_0$, $\delta x_N$ pairs (Supplementary Eqs. 36–41) resulting in the theoretical probability distribution $\sim \exp(-R_{min}/k_BT)$ described by the parameters $\sigma_{x_0}$, $\sigma_{x_N}$, and $\rho$ (Supplementary Eqs. 42–44).

**Modeling the reduction of threshold electric field.** The stably trapped conformation of the molecule was determined as described above, providing the minimum effective free energy at a given electric field strength. As the molecule is displaced from this minimum position towards the nanochannel, the effective free energy increases and reaches a maximum at a position where a certain portion of the DNA molecule is inserted into the nanochannel. The maximum energy conformation at a given electric field is analogous to the transition state in chemical kinetics. The difference between the maximum and minimum effective free energies, which is the effective free-energy barrier, $\Delta F$, was used to calculate the mean residence time, $\tau$, of the molecule in the nanofunnel (Eq. 1).

**Comparison of theoretical and experimental values.** The theoretical predictions were fit to the experimental data by adjusting the relative contributions of the electrohydrodynamic, entropic, and osmotic gradient forces using a weighted least squares method. This was realized in practice by multiplying each of the electrohydrodynamic, entropic, and osmotic gradient contributions to the effective free energy calculated from the first principles by a coefficient of order unity. These three coefficients, considered as fitting parameters, were determined by simultaneously optimizing across the various λ-phage and T4-phage datasets: (1) the residence time measurements ($\tau$) of the various nanofunnels (Fig. 3a) and (2) the mean position measurements ($x_0$, $x_N$), and (3) fluctuation measurements

($\sigma_{x_0}$, $\sigma_{x_N}$, $\rho$) in the $\alpha = 0.45$ nanofunnel (Fig. 4a, d; Supplementary Fig. 5a, d). That is, the fit was performed across eighteen independent datasets. Each data point was weighted by the reciprocal of its variance and the sum of squared residuals was minimized using the steepest descent algorithm.

**Data availability.** Fluorescence microscopy images, the end positions of the molecules measured at each time point, and the MATLAB code used to produce the end positions are available from the authors upon request.

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

## Acknowledgements

This work was sponsored in part by grants from the National Human Genome Research Institute, National Institutes of Health (Grant Number R01HG002647 to J.M.R.). M.R. would like to acknowledge financial support from the National Science Foundation under Grants DMR-1309892, DMR-1436201, and DMR-1121107, the National Institutes of Health under Grants P01-HL108808 and 1UH2HL123645, and the Cystic Fibrosis Foundation. We are grateful to the staff of the Chapel Hill Analytical and Nanofabrication Laboratory (CHANL) for their support.

## Author contributions

J.Z., Y.W., L.D.M., S.P., M.R., and J.M.R. designed experiments. J.Z. and L.D.M. performed experiments and analyzed data. Y.W., S.P., and M.R. developed the theoretical model of nanofunnel-confined DNA molecules. Y.W. performed the calculations and fit the theoretical to experimal results. J.Z., Y.W., L.D.M., S.P., M.R., and J.M.R. wrote and edited the manuscript.

## Additional information

**Competing interests:** The authors declare no competing financial interests.

