## [Peer Review File · Nature Communications]

Reviewers' comments:

Reviewer #1 (Remarks to the Author):

The main idea of the work is to suggest a novel geometry for a nanopore through which DNA can be transported by the electric field. Authors show that fabrication of funnel-shaped nanopores is possible and that DNA motion through them is indeed observed. These observations have a good potential to find practical applications and to attract a wide readership, although this article is just the proof of principle.

The manuscript is nicely presented and is easy to read. The only presentation issue I have is with the figure 1, where all three forces are shown by arrows of different colors. I think this presentation is misleading. Electrohydrodynamic forces does indeed act on any monomer, wherever it is located, at any time, although the value of it may fluctuate. By contrast, two other forces, osmotic gradient and entropic force are both a rather more averaged mean field concepts. (By the way osmotic gradient is also an osmotic force if the solvent is athermal, which means the terminology in the paper is not optimal in this sense).

The attractive feature of the article is the combination of theory and experiment. In theoretical part, it is not clear to me how the results of this paper correspond to earlier results on a more traditional pore geometry, where tension propagation along the chain in the form of a so-called iso-flux trumpet) is known to describe the translocation dynamics. It seems that in the limit of very wide funnels authors should arrive at the same conclusion, but I do not see it. This is a potentially sensitive issue, because authors compute relevant barrier height from quasi-equilibrium considerations of the free energy, while tension propagation is a decisively non-equilibrium phenomenon.

I think that upon clarification of these issues the paper should be publishable in this journal.

Reviewer #2 (Remarks to the Author):

This is a very interesting and extremely well written paper and I support publication in Nature Communications with some modifications. The experiments describe the injection of DNA into a relatively wide nanochannel (> 100 nm) using a funnel that goes from roughly the radius of gyration down to the nanochannel side. The remarkable result of the paper is that the larger DNA is more easily injected into the channel, even though the confinement free energy cost is higher. The authors explain this through a very straightforward and well explained modeling effort (largely relegated to the SI, which does help with the length of the main text) that shows that osmotic forces due to non-uniform stretching are key to this non-intuitive result. The model that the authors develop has only 3 fitting parameters that need to be $O(1)$ and describes all the phenomenology in the experiments and is quite remarkable as a quantitative theory — all orders of magnitude are correctly predicted by the model. I was especially impressed by the agreement in Fig. S8; it's a shame that's not somehow incorporated into Fig. 4 but I understand the desire to show agreement across many parameters in that Figure. The main text is sufficiently compact for this journal, although I have some suggestions below for improvement. The figures are attractive and convey the important information. In summary, this is an very good paper.

My suggestions for revision are not especially long but I do encourage the authors to consider them in revising their manuscript:

=== MAIN TEXT ===

- The FIB method is great for this type of experiment in a lab. I'm not sure how easily this is going to scale up for the genomic applications envisioned by the authors. To get to human genomes, for example, is going to require massive parallelization that is not so obviously available with this method. I do not see this as an issue with the paper overall, as the science here is very strong, but perhaps the authors would like to mention the challenges (or lack thereof) to turn this into a genomics tool. I see this as a useful addition for a broad audience journal like Nature Communications. Overall, I thought the key contributions here are scientific and the technological impact is likely to be limited.

- Along the same lines, injection into 120 nm channels is not especially challenging. This can be done with pressure driven flow. The challenges always lie in injecting lots of molecules quickly so that the throughput is good. I do not see a need to address this particular point in the paper, but I wanted to raise it here in the review.

- It would be worthwhile for the authors to clarify the difference between their funnel and the method from Bob Austin's group (Cao et al., Appl. Phys. Lett. 81, 3058-3060, 2002). There are differences, and I do not see this as affecting the novelty of this paper, but it is good to put the new work in the context of the literature.

- Along the same lines, Reisner and coworkers have discussed the role of osmotic forces in confinement in a related problem of compressing DNA in Khorshid et al., Phys. Rev. Lett. 113, 268104 (2014) and Khorshid et al., Macromolecules 49, 1933-1940 (2016). It would be helpful to connect the present work to the literature and explain clearly what is new here.

- In the section on Image Analysis, the authors might want to clarify that the "molecule ends" are not literally the first base and last base of the molecule. I assume they are the most distal and most proximate bright spots with respect to the channel entrance. In these levels of confinement, the chain is coiled so the ends are not necessarily the distal segments. This is just a notational comment, not a criticism of the analysis.

- In the main text theoretical modeling, the authors cite the backbone diameter. I recommend they replace this (or augment it) with the effective width of 6 nm in the SI. For people familiar with this problem, the first thought is that the excluded volume is going to be too low, but that's not actually the case if one takes the time to read the rather lengthy SI to find this value.

- The authors are rather neutral in the text about the model. I think it's very good in almost all respects and they are underselling the result (although, given typical submission to high impact journals, the modesty is much appreciated). The model does seem to have some challenges in predicting the molecular weight dependence in Fig. 3. Can the authors provide some explanation for why this particular quantity is so much harder to capture than all of the other dependences (e.g., electric field or α).

=== SUPPORTING INFORMATION ===

- Please explain a bit more clearly why there is no stable trapping for $\alpha > 1.5$ on the first page.

- The sigmoidal approach in Fig. S4 is reasonable given the phenomenology for larger values of α . However, it somewhat undercuts the claim that the injection is improved by a factor of 30 fold. That value, presumably obtained from the asymptote in Fig. 3b, involves the larger α data. This concern is attenuated somewhat by the data for $\alpha = 0.78$, which also appears to be close to the 30 fold value without requiring a change in the analysis method.

- The paragraph following SI.13 points out that the analysis of the electrohydrodynamics is an extension of Ref. S12. Ref. S12 is an important paper in this field, and the extension of the result is (in my opinion) another important accomplishment of the present paper. It would be nice to have this sentence in the main text somewhere so that it does not get lost in the SI. Since many people do not read the SI, I am concerned that it will be overlooked.

Reviewer #3 (Remarks to the Author):

The work in review consists of understanding electric field-driven DNA transport dynamics in a nanofunnel that interfaces microscale and nanoscale confinement. A nanofunnel geometry has been proposed with the aim of achieving a reduction in the free-energy requirement or threshold electric field that is associated with the lowering of conformational entropy due to gradual confinement. It has also been demonstrated that nanofunnels can act as stable DNA traps. Both, experimental observations and theoretical modeling of the physics involved, were presented.

Over the years, the DNA transportation from open space, microscale to nanoscale environment has been well studied, and there are even commercial products based on nano-channel. An exhaustive review of previous work done to gradually confine DNA molecules from microscale environment to nanochannels is necessary for comparing the presented work. Mentioned below are few publications that are relevant to the manuscript.

Cao et al. (2002) have reported a gradient micro/nano-post structure to lower the entropic barrier for DNA molecules into nanochannels. This model is tuned towards high-throughput DNA stretching in nanochannel arrays.

Following this, Randall et al. (2006), have published a method for electric field-gradient stretching of DNA in a funneled-microchannel. A funnel geometry was utilized to create an electric-field gradient much like the work in review.

Wu et al. (2011) demonstrated the fabrication of a plastic nanofluidic device that incorporates a funnel interface for efficient DNA entry (lower threshold electric field and conformational entropy), from a 30 μm to 75 nm channel. In contrast, the proposed nanofunnel scales from 1.2 μm to 120 nm.

Specific comments:

1. The authors failed to acknowledge the free energy barrier from open space (DNA loading port) to microscale confinement is even more problematic. This is evident that 120v was employed to drive the DNA to 1.6 micron funnels.
2. It has also been suggested that nanofunnels may potentially be used as a force spectroscopy tool, without sufficient perspective on how this would compare to other force spectroscopy tools.
3. The observation that longer DNA molecules can more readily enter nanochannels via a nanofunnel is an important outcome of the study. It will be interesting to see how this finding can be translated into megabase-long DNA molecules both experimentally and theoretically. This is more relevant in nanochannel applications.
4. Following the point 3, it would be nice to provide data point of loading mixed DNA samples of different length into the nanochannel further the claim.

References:

1. Cao, Han, et al. "Gradient nanostructures for interfacing microfluidics and nanofluidics." *Applied Physics Letters* 81.16 (2002): 3058-3060.

2. Wu, Jiahao, et al. "Complete plastic nanofluidic devices for DNA analysis via direct imprinting with polymer stamps." *Lab on a Chip* 11.17 (2011): 2984-2989.
 3. Randall, Greg C., Kelly M. Schultz, and Patrick S. Doyle. "Methods to electrophoretically stretch DNA: microcontractions, gels, and hybrid gel-microcontraction devices." *Lab on a Chip* 6.4 (2006): 516-525.
- A. Liu, Xu, Mirna Mihovilovic Skanata, and Derek Stein. "Entropic cages for trapping DNA near a nanopore." *Nature communications* 6 (2015).

Authors' Replies to Reviewers' comments

Reviewer #1: The main idea of the work is to suggest a novel geometry for a nanopore through which DNA can be transported by the electric field. Authors show that fabrication of funnel-shaped nanopores is possible and that DNA motion through them is indeed observed. These observations have a good potential to find practical applications and to attract a wide readership, although this article is just the proof of principle. The manuscript is nicely presented and is easy to read.

Authors' reply: We thank Reviewer #1 for very complementary evaluation of our work and for the comments that allowed us to further improve the quality of our paper.

Reviewer #1 comment 1: The only presentation issue I have is with the figure 1, where all three forces are shown by arrows of different colors. I think this presentation is misleading. Electrohydrodynamic forces does indeed act on any monomer, wherever it is located, at any time, although the value of it may fluctuate. By contrast, two other forces, osmotic gradient and entropic force are both a rather more averaged mean field concepts. (By the way osmotic gradient is also an osmotic force if the solvent is athermal, which means the terminology in the paper is not optimal in this sense).

Authors' reply 1.1: We thank reviewer for this comment. The reviewer is correct that osmotic gradient and entropic forces are directly applied onto different monomers at different moments of time. Our convention indeed corresponds to pre-averaging DNA concentration fluctuation and spreading these forces over all monomers with the same coordinate x . We add this clarification to the text (page 4, lines 79-86):

“The contributions of these forces are shown schematically in Figure 1, where the arrows of different colors suggest an average force over all monomers at a particular location within the nanofunnel. We note that the osmotic gradient and entropic forces are acting only on monomers in contact with each other or with the nanofunnel wall. As the DNA molecule fluctuates, these forces are directly applied onto different monomers at different moments in time. The concentration fluctuations are suppressed within the mean field theory, and the DNA configuration is completely characterized by the forces pre-averaged over all monomers at a location along the nanofunnel's longitudinal axis.”

Reviewer #1 comment 2: The attractive feature of the article is the combination of theory and experiment. In theoretical part, it is not clear to me how the results of this paper correspond to earlier results on a more traditional pore geometry, where tension propagation along the chain in the form of a so-called iso-flux trumpet) is known to describe the translocation dynamics. It seems that in the limit of very wide funnels authors should arrive at the same conclusion, but I do not see it. This is a potentially sensitive issue, because authors compute relevant barrier height from quasi-equilibrium considerations of the free energy, while tension propagation is a decisively non-equilibrium phenomenon.

Authors' reply 1.2: In Fig. 3 we present our results for different funnel geometries and, in particular, for traditional no funnel or vestibule geometries. In these latter two cases, our theory is in reasonable agreement with trapping times calculated earlier, in newly added Ref. 26 [Han, J.; Turner, S. W.; Craighead, H. G. Entropic trapping and escape of long DNA molecules at submicron size constriction. *Phys. Rev. Lett.* **83**, 1688-1691 (1999)]. As reviewer #1 noticed, in this paper we do not study any non-equilibrium translocation dynamics and, therefore, cannot reproduce non-equilibrium effects of tension propagation

along the chain. On page 7 (lines 142-151) we expand on the discussion of our results in the context of prior work:

“The transition from the $\alpha = 0$ nanofunnel to the nanochannel is similar to an entropic trap such as those reported by Han et al., which consisted of 30- μm wide channels with alternating deep (1.4 μm) and shallow (90 nm) segments.²⁵ The critical dimensions of our $\alpha = 0$ nanofunnel are similar, but the reduction in both the width and depth dimensions in our case presents both a larger entropic barrier and a smaller collisional cross-section between the DNA molecules and the nanochannel entrance. It is therefore unsurprising that the residence times directly measured at the nanochannel entrance are longer (by an order of magnitude) than those calculated for comparable electric fields by Han et al. from mobility data in entropic trap devices. As in our experiments, Han et al. observed shorter residence times for longer DNA molecules, an effect that, when multiplied over thousands of entropic traps in series, enabled size-dependent DNA separations.²⁶”

Reviewer #1: I think that upon clarification of these issues the paper should be publishable in this journal.

Authors' reply: We believe that we have satisfactorily replied to Reviewer's comments and added the corresponding clarifications to our paper.

Reviewer #2: This is a very interesting and extremely well written paper and I support publication in Nature Communications with some modifications. The experiments describe the injection of DNA into a relatively wide nanochannel (> 100 nm) using a funnel that goes from roughly the radius of gyration down to the nanochannel side. The remarkable result of the paper is that the larger DNA is more easily injected into the channel, even though the confinement free energy cost is higher. The authors explain this through a very straightforward and well explained modeling effort (largely relegated to the SI, which does help with the length of the main text) that shows that osmotic forces due to non-uniform stretching are key to this non-intuitive result. The model that the authors develop has only 3 fitting parameters that need to be $O(1)$ and describes all the phenomenology in the experiments and is quite remarkable as a quantitative theory — all orders of magnitude are correctly predicted by the model. I was especially impressed by the agreement in Fig. S8; it's a shame that's not somehow incorporated into Fig. 4 but I understand the desire to show agreement across many parameters in that Figure. The main text is sufficiently compact for this journal, although I have some suggestions below for improvement. The figures are attractive and convey the important information. In summary, this is a very good paper. My suggestions for revision are not especially long but I do encourage the authors to consider them in revising their manuscript:

Authors' reply: We thank Reviewer #2 for highly positive evaluation of our work and for his/hers constructive and detailed comments and suggestions for improvement of our paper. We tried to incorporate the contents of Figure S8 into Figure 4 upon the reviewer's recommendation but ultimately felt that this attempted to communicate too much information in the figure at the expense of clarity.

Reviewer #2 comment 1: The FIB method is great for this type of experiment in a lab. I'm not sure how easily this is going to scale up for the genomic applications envisioned by the authors. To get to human

genomes, for example, is going to require massive parallelization that is not so obviously available with this method. I do not see this as an issue with the paper overall, as the science here is very strong, but perhaps the authors would like to mention the challenges (or lack thereof) to turn this into a genomics tool. I see this as a useful addition for a broad audience journal like Nature Communications. Overall, I thought the key contributions here are scientific and the technological impact is likely to be limited.

Authors' reply 2.1: We thank Reviewer #2 for the suggestion and have added some remarks to this point in the concluding paragraph of the paper (page 10, lines 220-224). We agree that such a comment helps to address the technological merit of the work.

“While the FIB milling fabrication method used here is admittedly a low throughput technique, the flexibility of generating highly useful three-dimensional nanofunnels motivates its use, especially in conjunction with higher throughput fabrication methods. FIB milling could be used, for example, to fabricate masters for the subsequent fabrication of many fluidic devices through molding or embossing methods.¹¹”

Reviewer #2 comment 2: Along the same lines, injection into 120 nm channels is not especially challenging. This can be done with pressure driven flow. The challenges always lie in injecting lots of molecules quickly so that the throughput is good. I do not see a need to address this particular point in the paper, but I wanted to raise it here in the review.

Authors' reply 2.2: We thank Reviewer #2 for raising this important point.

Reviewer #2 comment 3: It would be worthwhile for the authors to clarify the difference between their funnel and the method from Bob Austin's group (Cao et al., Appl. Phys. Lett. 81, 3058-3060, 2002). There are differences, and I do not see this as affecting the novelty of this paper, but it is good to put the new work in the context of the literature.

Authors' reply 2.3: We thank Reviewer #2 for this suggestion and have expanded the introductory section to encompass this work (page 3, lines 47-52).

“For example, gradient structures consisting of an array of posts aligned to a nanochannel array were described by Cao et al., where the inter-post distance and the channel depth decrease from the DNA reservoir to the nanochannel entrances.⁸ Multiple pathways of gradually increasing confinement are thus provided to a molecule. While these structures are demonstrably useful for facilitating DNA entry into the nanochannels, their multi-path nature complicates the study of DNA behavior, which has not been modeled therein.”

Reviewer #2 comment 4: Along the same lines, Reisner and coworkers have discussed the role of osmotic forces in confinement in a related problem of compressing DNA in Khorshid et al., Phys. Rev. Lett. 113, 268104 (2014) and Khorshid et al., Macromolecules 49, 1933–1940 (2016). It would be helpful to connect the present work to the literature and explain clearly what is new here.

Authors' reply 2.4: We agree that there is a relevant overlap between our work and that of Reisner. We have added a detailed discussion of this work on page 12 of the SI:

“We note that the monomer concentration profiles shown in Supplementary Fig. 7 appear similar to those of molecules compressed against a nanosphere held by optical tweezers within a nanochannel.^{517,518} While the elegant experiments of Khorshid et al. offer another force measurement methodology, there are key differences with our work that should be emphasized. First, our work exclusively explores DNA molecules at quasi-equilibrium whereas the measurements of Khorshid et al. (most importantly, the ones that yield concentration profiles like those of the present work) are performed on transient conformations or steady-state conformations far from equilibrium. Second, the presence of a nanosphere that largely occludes the nanochannel significantly impacts fluid flow in the system of Khorshid et al. Third, while it is our opinion that both methods can yield important information about near equilibrium and dynamic behavior of large polyelectrolytes, the nanofunnels are more directly applicable to nanofluidic devices for the analysis of biological macromolecules given their ability to lower the threshold electric field needed to induce DNA entry into the nanochannels.”

We have also briefly mention this work on page 9 (lines 195-197) of the main text under the Results and Discussion section:

“Previously reported compression of DNA molecules confined within an nanochannel against an obstruction or nanobead is analogous to compression within an $\alpha = 0$ nanofunnel.³¹⁻³³”

Reviewer #2 comment 5: In the section on Image Analysis, the authors might want to clarify that the “molecule ends” are not literally the first base and last base of the molecule. I assume they are the most distal and most proximate bright spots with respect to the channel entrance. In these levels of confinement, the chain is coiled so the ends are not necessarily the distal segments. This is just a notational comment, not a criticism of the analysis.

Authors’ reply 2.5: Reviewer is correct that “molecule ends” experimentally correspond the most distal and proximate bright spots. Theoretically, we use scaling (blob) model of DNA in a funnel/channel in which DNA end positions are localized only down to the size of the corresponding blobs. To this accuracy, the positions of DNA ends coincide with the positions of the first and last bases of the molecule. We have now commented on this in the Methods section on page 14 (lines 301-305):

“These ends experimentally correspond to the most distal and most proximal bright spots in the image and not exactly to the first and last base pairs of the molecule. The end positions in the scaling model are localized only down to the size of the corresponding “blobs”.³⁸ It is therefore to this level of accuracy that the measured end positions coincide with the positions of the corresponding terminal bases of the molecule.”

Reviewer #2 comment 6: In the main text theoretical modeling, the authors cite the backbone diameter. I recommend they replace this (or augment it) with the effective width of 6 nm in the SI. For people familiar with this problem, the first thought is that the excluded volume is going to be too low, but that’s not actually the case if one takes the time to read the rather lengthy SI to find this value.

Authors’ reply 2.6: We thank reviewer #2 for this suggestion and replace “backbone diameter” by the “effective DNA width” in the main text theoretical section on page 15 (lines 320-322) and referred to the work of Stigter.

“DNA was modeled as a semi-flexible chain having a persistence length of 50 nm and an effective DNA width of 6 nm that includes the contribution of the double layer formed by counterions around the backbone.³⁹”

Reviewer #2 comment 7.1: The authors are rather neutral in the text about the model. I think it’s very good in almost all respects and they are underselling the result (although, given typical submission to high impact journals, the modesty is much appreciated).

Authors’ reply 2.7.1: We thank reviewer #2 for highly complementary comments about our theoretical model. We believe that following the recommendations of the reviewers has resulted in a manuscript that better places our work in the context of the prior work and better communicates its significance. We have also added a sentence at the end of the abstract to draw more attention to the model, which, as the reviewer points out, was not highly touted in the main manuscript (page 1, lines 29-31).

“These technological applications illustrate the benefit of finely tuned nanoscale conduit geometries, which can be designed using the theoretical model developed here.”

Reviewer #2 comment 7.2: The model does seem to have some challenges in predicting the molecular weight dependence in Fig. 3. Can the authors provide some explanation for why this particular quantity is so much harder to capture than all of the other dependences (e.g., electric field or alpha).

Authors’ reply 2.7.2: Our theory contains three adjustable parameters in the expression of the free energy – 3 coefficients on the order of unity. We find the values of these coefficients from the comparison of our theoretical predictions with all available experimental data. Since only several sub-sets of data in Fig. 3 deal with molecular weight dependence, this dependence is less well captured and is not optimized as well by our choice of three adjustable parameters in comparison to more representative electric field and alpha dependences. In principle, these three configuration-dependent coefficients can be calculated from the first principles by accounting for the radial dependence of monomer density and other more detailed features of DNA configuration in the funnel, but this type of calculation is much more complicated and is well outside the scope of the paper.

Reviewer #2 comment 8: Please explain a bit more clearly why there is no stable trapping for alpha > 1.5 on the first page.

Authors’ reply 2.8: We added an explanation of why there is no stable trapping for alpha > 1.5 on page 1 of the Supporting Information.

“This is because the depth of the minimum of the effective free energy decreases with increasing alpha due to rapid variation of the diameter along such a funnel. At $\alpha > 1.5$ the well depth becomes smaller than $k_B T$ and thermal fluctuations are sufficient for a DNA molecule to escape the trap. The direction of this escape (towards the narrow end or wide end of the nanofunnel) is highly sensitive to the electric field in the nanofunnel...”

Reviewer #2 comment 9: The sigmoidal approach in Fig. S4 is reasonable given the phenomenology for larger values of alpha. However, it somewhat undercuts the claim that the injection is improved by a factor

of 30 fold. That value, presumably obtained from the asymptote in Fig. 3b, involves the larger alpha data. This concern is attenuated somewhat by the data for alpha = 0.78, which also appears to be close to the 30 fold value without requiring a change in the analysis method.

Authors' reply 2.9: We appreciate the reviewer's comments and agree that this bears further emphasis. We have added the following text to page 3 of the SI.

"We further note that one of the primary conclusions of this study — that the threshold electric field needed to initiate nanochannel entry is reduced 30-fold — is indicated not only by the $\alpha=1.46$ and $\alpha=1.89$ results but also by the residence time measurements conducted in the $\alpha=0.78$ nanofunnel. That is, the asymptotic behavior occurring at larger values of α (as the strength of the electro-osmotic piston is saturated) is spanned by multiple data sets."

And note in the conclusion on page 10 (lines 208-209) that:

"Given the rapid decrease in relative threshold electric field strengths as a function of α , nanofunnels with $\alpha=0.78$ were nearly as effective."

Reviewer #2 comment 10: The paragraph following SI.13 points out that the analysis of the electrohydrodynamics is an extension of Ref. S12. Ref. S12 is an important paper in this field, and the extension of the result is (in my opinion) another important accomplishment of the present paper. It would be nice to have this sentence in the main text somewhere so that it does not get lost in the SI. Since many people do not read the SI, I am concerned that it will be overlooked.

Authors' reply 2.10: We thank reviewer #2 for pointing out the novelty and importance of our electrohydrodynamic analysis. We follow Reviewer's advice and add a sentence about this theoretical model to main text on page 5 (lines 106-109):

"In addition to the identification of the electro-osmotic piston, a significant outcome of these modeling efforts is an extension of the important work by Long, Viovy, and Ajdari to describe the action of electric fields and non-electric forces on DNA molecules in confined environments.²²"

And add on page 14 (lines 326-329) in the Methods:

"In the Supplementary Information, we generalize an analysis of electrohydrodynamic forces given in Ref. 22 to the case of DNA trapped inside a nanofunnel, the walls of which suppress the fluid backflow around the molecule. In this case, the backflow can circulate only on microscopic scales through the effective pores inside DNA, similar to the case of a charged gel."

Reviewer #3: The work in review consists of understanding electric field-driven DNA transport dynamics in a nanofunnel that interfaces microscale and nanoscale confinement. A nanofunnel geometry has been proposed with the aim of achieving a reduction in the free-energy requirement or threshold electric field that is associated with the lowering of conformational entropy due to gradual confinement. It has also been demonstrated that nanofunnels can act as stable DNA traps. Both, experimental observations and theoretical modeling of the physics involved, were presented. Over the years, the DNA transportation from open space, microscale to nanoscale environment has been well studied, and there are even commercial products based on nano-channel. An exhaustive review of previous work done to gradually

confine DNA molecules from microscale environment to nanochannels is necessary for comparing the presented work. Mentioned below are few publications that are relevant to the manuscript.

Authors' reply: We thank Reviewer #3 for suggesting to enhance the review of prior works. Following suggestions of Review #3 we expanded the overview of prior works and comparison of these works to our results.

Reviewer #3 comment 1: Cao et al. (2002) have reported a gradient micro/nano-post structure to lower the entropic barrier for DNA molecules into nanochannels. This model is tuned towards high-throughput DNA stretching in nanochannel arrays.

Authors' reply 3.1: We thank Reviewer #3 for this suggestion and have expanded the introductory section to encompass this work on page 3 (lines 47-52).

“For example, gradient structures consisting of an array of posts aligned to a nanochannel array were described by Cao et al., where the inter-post distance and the channel depth decrease from the DNA reservoir to the nanochannel entrances.⁸ Multiple pathways of gradually increasing confinement are thus provided to a molecule. While these structures are demonstrably useful for facilitating DNA entry into the nanochannels, their multi-path nature complicates the study of DNA behavior, which has not been modeled therein.”

Reviewer #3 comment 2: Following this, Randall et al. (2006), have published a method for electric field-gradient stretching of DNA in a funneled-microchannel. A funnel geometry was utilized to create an electric-field gradient much like the work in review.

Authors' reply 3.2: We thank the reviewer for this comment and have cited the work of Randall on page 8 (lines 171-173) of the manuscript:

“We note that in these quasi-equilibrium measurements, the DNA molecules are extended by confinement and not as a result of any significant strain rate imposed by the electric field gradient.²⁷”

Reviewer #3 comment 3: Wu et al. (2011) demonstrated the fabrication of a plastic nanofluidic device that incorporates a funnel interface for efficient DNA entry (lower threshold electric field and conformational entropy), from a 30 μm to 75 nm channel. In contrast, the proposed nanofunnel scales from 1.2 μm to 120 nm.

Authors' reply 3.3: We thank the reviewer to bringing this work to our attention. We have cited this work and provided a description in the introductory paragraphs on page 3 (lines 57-62):

“We note that a FIB milled “funnel-like inlet,” consisting of a series of discrete reductions in conduit width and depth, has previously been reported to assist with DNA entry into a nanochannel, although the effect was not quantified.¹¹ In the present study, the gradual and smooth transition from microscale to nanoscale confinement within our FIB milled nanofunnels, in contrast to a coarser transitioning of DNA confinement in a stepwise fashion, is an important enabling aspect of both the experimental measurements and modeling efforts.”

Reviewer #3 comment 4: The authors failed to acknowledge the free energy barrier from open space (DNA loading port) to microscale confinement is even more problematic. This is evident that 120v was employed to drive the DNA to 1.6 micron funnels.

Authors' reply 3.4: We respectfully disagree with Reviewer #3 on this point. There is no significant free energy barrier from the reservoirs to the microscale confinement as the nanofunnel diameter at its wide end is on the order of the radii of gyration (page 5). To avoid confusion, we have added a note of clarification on the use of a high field strength as a matter of convenience to introduce the DNA molecules to the nanofunnel on page 12, line 271 – page 13, line 272.

“The DNA concentrations used were sufficiently low that occupancy of two molecules within a nanofunnel during a measurement was rare. Introduction of a molecule to the nanofunnel was therefore expedited by using a relatively high (1500 V/cm) nanochannel electric field to draw in a molecule from the source microfluidic channel (corresponding to an electric field at the nanofunnel mouth of 5-10 V/cm, depending on nanofunnel shape).”

Reviewer #3 comment 5: It has also been suggested that nanofunnels may potentially be used as a force spectroscopy tool, without sufficient perspective on how this would compare to other force spectroscopy tools.

Authors' reply 3.5: We appreciate the Reviewer's feedback and have added a comparison of the proposed technique with other force spectroscopy tools in the Results and Discussion section on page 9 (lines 188-202):

“Each of the above parameters determined by imaging a stably trapped DNA molecule are sensitive to the length, linear charge density, effective width, and persistence length of the molecule (Supporting Information). Electro-osmotic tweezers can therefore be used to measure changes to these characteristics that are produced, for example, by changes to the solution ionic strength.²⁸ We note that the use of multiple complementary force spectroscopies is useful for revealing subtle differences in polymer dynamical behavior.²⁹ While DNA stretching and twisting behavior has been extensively studied using atomic force microscopy, optical tweezers, and magnetic tweezers, far fewer measurements of DNA compression have been reported due to a lack of available tools.³⁰ Previously reported compression of DNA molecules confined within a nanochannel against an obstruction or nanobead is analogous to compression within an $\alpha = 0$ nanofunnel.³¹⁻³³ In the present study, however, we demonstrate electro-osmotic tweezers that are tunable through the nanofunnel dimensions and geometry, promising the availability of measurements to various charged macromolecules across different force regimes. In addition to complementing existing stretching and twisting force spectroscopies, the ability to measure compression is relevant to better understanding phenomena such as nucleic acid packaging (e.g., in viral capsids).³⁴”

Reviewer #3 comment 6: The observation that longer DNA molecules can more readily enter nanochannels via a nanofunnel is an important outcome of the study. It will be interesting to see how this

finding can be translated into megabase-long DNA molecules both experimentally and theoretically. This is more relevant in nanochannel applications.

Reviewer #3 comment 7: Following the point 3, it would be nice to provide data point of loading mixed DNA samples of different length into the nanochannel further the claim.

Authors' reply 3.6 and 3.7: We absolutely agree with the reviewer that the extension of this work to longer (Mbp-long) DNA molecules is particularly relevant to nanochannel applications. We are interested in expanding the range of DNA sizes to this length scale and exploring, as the reviewer suggests. However, we believe that that is beyond the scope of the present work. It entails overcoming some practical difficulties in handling such long DNA molecules gently enough prior to their addition to the fluidic device to avoid shearing. Monodisperse samples of Mbp-long DNA are not available, which is the reason that most work in the field has been done using λ -phage, T4-phage, or T2-phage DNA (see e.g., Table 3 of Dorfman et al. Chem. Reviews 113, 2584-2667). We envision needing to incorporate an accurate sizing protocol to determine the contour length of each DNA molecule that is interrogated.

REVIEWERS' COMMENTS:

Reviewer #1 (Remarks to the Author):

I think the revised version of the paper addressed satisfactorily all critical remarks and in my opinion the paper is good to go. I enthusiastically recommend acceptance and publication.

Reviewer #2 (Remarks to the Author):

The response from the authors is satisfactory. I recommend publication.

Reviewer #3 (Remarks to the Author):

The authors have adequately addressed the reviewer's comments. The paper can be published as it is.